# Predictive Model of the Relationship between Appearance, Eating Attitudes, and Physical Activity Behavior in Young People amid COVID-19

**DOI:** 10.3390/nu16132065

**Published:** 2024-06-28

**Authors:** Jianye Li, Dominika Wilczynska, Małgorzata Lipowska, Ariadna Beata Łada-Maśko, Bartosz M. Radtke, Urszula Sajewicz-Radtke, Bernadetta Izydorczyk, Taofeng Liu, Zitong Wang, Junyu Lu, Mariusz Lipowski

**Affiliations:** 1Faculty of Physical Education, Gdańsk University of Physical Education and Sport, Górskiego 1, 80-336 Gdansk, Poland; dominika.wilczynska@awf.gda.pl (D.W.); zitong.wang@awf.gda.pl (Z.W.); junyu.lu@awf.gda.pl (J.L.); 2Institute of Psychology, University of Gdańsk, Bażyńskiego 8 Street, 80-309 Gdansk, Poland; malgorzata.lipowska@ug.edu.pl (M.L.);; 3Laboratory of Psychological and Educational Tests, 80-239 Gdansk, Polandsajewicz-radtke@pracowniatestow.pl (U.S.-R.); 4Institute of Psychology, Jagiellonian University, 30-060 Krakow, Poland; bernadetta.izydorczyk@uj.edu.pl; 5Physical Education Institute (Main Campus), Zhengzhou University, Zhengzhou 450001, China; liutaofeng@zzu.edu.cn; 6Faculty of Social and Humanities, WSB Merito University Gdansk, 80-226 Gdańsk, Poland; mariusz.lipowski@gdansk.merito.pl

**Keywords:** negative image fear, eating disorders, physical activity targeting, body image anxiety, COVID-19

## Abstract

This cross-sectional study conducted in Poland explored the relationship between the fear of negative appearance evaluations, eating disorders, and physical activity objectives, particularly during the COVID-19 pandemic. The Fear of Negative Appearance Evaluation Scale (FNAES), the Eating Attitude Test (EAT-26), and the Physical Activity Goals Inventory (IPAO) were administered to 644 participants (455 males with a mean age of 35.2 ± 6.2 years and 189 females with a mean age of 30.18 ± 5.7 years). This study explored the effects of gender, age, and body mass index (BMI) on FNAES, EAT-26, and IPAO scores. The results of this study demonstrated that females scored higher on fear of negative appearance, peaking at 41–50 years of age. Distinct BMI categories were associated with different negative appearance fear scores, eating attitudes, and physical activity objectives. Significant correlations were also found between the fear of negative appearance, dietary attitudes, and physical activity goals. Eating attitudes completely moderated the relationship between the fear of negative appearance and physical activity objectives. A significant interaction effect of age and body mass index on physical activity objectives was also revealed. These results highlight the relevance of considering gender, age, and body mass index when examining the associations between the fear of negative appearance, eating attitudes, and physical activity objectives.

## 1. Introduction

The COVID-19 epidemic was first discovered in Wuhan, China in 2019, and it has aroused widespread concern around the world. The World Health Organization (WHO) declared COVID-19 as a pandemic on 11 March 2020. Social isolation was adopted as a strategy to reduce the transmission speed of the SARS-CoV-2 virus in several countries around the world [1]. After the onset of the pandemic, economic and sports activities in many countries were restricted, and with the new coronavirus epidemic, people’s daily behaviors, including physical activities, were disrupted, potentially affecting their dietary and body image management [2], thereby exacerbating the fear of a negative appearance, which involves individuals’ anxiety and apprehension about how their physical features are perceived [3], manifesting as a deep concern about being negatively evaluated or judged because of one’s appearance [4]. The fear of a negative appearance evaluation refers to the anxiety and apprehension individuals experience concerning how their physical features are perceived and judged by others. This fear often manifests as a deep concern about being evaluated negatively or being seen as unattractive, which can lead to significant social anxiety and influence various aspects of one’s behavior and self-esteem [5,6,7].

This fear often leads to individuals experiencing high levels of social anxiety, especially if they feel that their appearance is being scrutinized, and is significantly associated with eating attitudes and mood [8]. It may also trigger body image issues, leading individuals to adopt unhealthy behaviors, such as restrictive eating habits or excessive exercise, in an attempt to change their appearance to meet social standards [9,10]. A study of Italian adolescents concluded that an increase in obesity indicators (BMI, waist circumference, waist-to-height ratio) and weight due to reduced physical activity leads to dissatisfaction with body image [11]. The fear of negative appearance refers to individuals’ anxiety and apprehension about how their physical features are perceived, which can lead to body dissatisfaction and maladaptive eating behaviors. Studies have shown a significant correlation between the fear of negative appearance and eating attitudes [12,13]. People often start with emotional eating before reaching an extreme eating disorder, especially when the overall context is a crisis like a global pandemic [14].

In addition, the fear of negative appearance may be influenced by sociocultural factors, with different cultural norms and expectations shaping individuals’ perceptions of beauty and attractiveness. Overall, the fear of negative appearance can significantly impact an individual’s mental health and may lead to the development of a variety of mental health problems, including eating disorders and mood disorders [15]. A growing body of research has demonstrated significant associations among one’s appearance and eating and physical behavioral activities [16,17].

Due to concerns about physical appearance, it is not surprising that people will try to obtain a desirable physical appearance by being physically active or restricting their diets to alleviate appearance-related anxiety. On the other hand, chronic eating irregularities can lead individuals to develop symptoms of eating disorders, which are defined as abnormal eating habits and behaviors influenced primarily by psychosomatic factors [18]. It is a problem that exists worldwide and is particularly pronounced in young women, with the main clinical manifestations being bulimia and anorexia nervosa [19].

Eating disorders among adults often arise from multifaceted etiological factors, encompassing psychological, sociocultural, and biological determinants. Central to the pathology of these disorders is a distorted body image, intricately linked with the apprehension surrounding negative appearance evaluation. This dissatisfaction commonly emanates from societal pressures and unrealistic beauty standards that media representations and cultural paradigms have propagated. As individuals internalize these standards, they may develop an intense fear of adverse judgments based on their appearance. This fear often instigates maladaptive behaviors, including restrictive dietary practices, compulsive exercising, or purging, in pursuit of achieving an idealized body image [20].

The apprehension of negative appearance serves to exacerbate body dissatisfaction, establishing a deleterious cycle wherein individuals feel compelled to engage in harmful behaviors to attain an unattainable aesthetic ideal. Such behaviors frequently precipitate the onset or exacerbation of eating disorders, such as anorexia nervosa, bulimia nervosa, or binge eating disorder. Moreover, the fear of negative appearance may function as a perpetuating factor for these disorders, reinforcing disordered eating behaviors, even following the initiation of treatment [21]. Individuals may persist in maladaptive dietary habits as they endeavor to mitigate anxieties regarding their appearance, thereby further entrenching their negative body image perceptions.

Eating behaviors, influenced by the fear of negative appearance, and how they can impact individuals’ physical activity objectives have been investigated in several studies [22,23]. There is a suggested association between physical activity and eating to a certain degree. A study of Chinese university students has indicated that individuals with eating disorders exhibit abnormal levels of exercise intensity, commonly referred to as “exercise dependence”, which is particularly prevalent among patients with anorexia nervosa [24,25]. Other studies also confirm the findings that exercise dependence can be harmful to health and can also affect eating behavior and cause eating disorders. Compulsive exercise is a common symptom in patients with eating disorders [26,27]. The main goal of excessive exercise symptoms in patients with eating disorders is to achieve weight loss and physical fitness through excessive exercise intensity. From this, it can be assumed that eating disorder symptoms can influence a person’s physical behavior and activity goals.

In the context of recent research, several studies have explored the intricate relationships among physical appearance, dietary behavior, and engagement in physical activity, particularly focusing on young populations. Randomized controlled trials (RCTs) and quasi-experimental designs with control groups have shed light on these dynamics, offering insights into the underlying mechanisms that drive behavior in these domains. Recent RCTs and quasi-experimental studies have underscored the significant impact of sociocultural factors, media influences, and societal beauty standards on individuals’ perceptions of their physical appearance [28]. These studies highlight how exposure to idealized body images in media platforms can contribute to body dissatisfaction and the development of eating disorders among young people [29]. Furthermore, they emphasize the role of social comparison processes in shaping individuals’ attitudes toward their bodies, with comparisons often leading to negative self-evaluations and maladaptive behaviors [30].

Moreover, theoretical frameworks such as Social Comparison Theory and Objectification Theory provide a robust foundation for understanding the relationship among physical appearance, eating disorders, and motives behind engagement in physical activity [31]. Social Comparison Theory posits that individuals tend to evaluate their own appearance and abilities based on comparisons with others, particularly those deemed as socially relevant or aspirational [32]. This process can fuel feelings of inadequacy and drive efforts to conform to societal beauty ideals, thereby influencing dietary behaviors and exercise patterns.

Objectification Theory, on the other hand, elucidates how societal objectification of bodies, especially those of young women, can lead to self-objectification and body surveillance [33]. This heightened self-awareness of one’s appearance can foster body dissatisfaction and contribute to disordered eating behaviors as individuals strive to attain unattainable beauty standards [34].

In the proposed structural regression model, these theoretical frameworks provide a conceptual basis for understanding the pathways through which the fear of negative appearance influences eating attitudes and, subsequently, physical activity objectives. By integrating these theories into the model, we aim to elucidate the underlying mechanisms driving behavior in these domains and provide valuable insights for intervention and prevention efforts targeting eating disorders and promoting healthy body image and physical activity behaviors among young populations. By addressing these significant gaps and extending previous findings, the current study offers valuable contributions to the understanding of how appearance-related fears influence eating behaviors and physical activity goals, emphasizing the importance of considering demographic variables in health promotion strategies.

Based on the literature read and summarized by previous related researchers, this study hypothesized that the effect of the fear of negative appearance on physical activity objectives involves a total of two stages. The first stage is that the attitude of fear of negative appearance will influence eating behaviors. The second stage is that eating behaviors will eventually influence people’s physical behavior and activity goals. Therefore, this study proceeds with the following hypotheses: (1) there is a significant correlation among the three factors, (2) there is a full mediating effect of fear of negative appearance between eating attitudes and physical activity objectives, and (3) physical activity objectives can be affected by different age stages and BMI groups.

## 2. Materials and Methods

### 2.1. Research Design

A quantitative, correlational, and descriptive cross-sectional study was conducted [35]. The data used in this study were part of a large international research project registered in the Protocol Registration and Outcomes System (ClinicalTrials.gov; https://clinicaltrials.gov/ct2/show/NCT04432038, accessed on 1 April 2020). The research team began by providing qualified researchers with research procedures and ethics training. An informed consent form and an online address for completing the questionnaire were also e-mailed and sent via WhatsApp and Facebook to those who met the inclusion criteria. At the same time, the larger population that met the inclusion criteria was asked to help invite friends and family to participate (“snowball sampling technique”). All survey participants provided informed consent and questionnaires. The questionnaire was translated into Polish and passed reliability and validity using KMO and Bartlett’s test.

### 2.2. Participants and Procedure

The investigation engaged 644 adult citizens from Poland, comprising 455 males and 189 females, all characterized by good health and an absence of disabilities. Participants had an average age of M = 32.69, with an age range spanning 18 to 74 years (standard deviation [SD] = 11.52 years).

The selection of groups was conducted through purposeful sampling. Participants were recruited using convenience and purposive sampling methods. The following inclusion criteria were used: age over 18 years, Polish nationality, no physical disability or physical disease preventing physical activity, and not receiving any treatment for eating disorders. These criteria were validated with the help of a questionnaire, which allowed for the identification of exclusion factors. An analysis of efficacy by F-test with G-power software 3.1 version indicated that the minimum sample size to produce a statistical efficacy of at least 0.9, an alpha value of 0.05, and a medium effect size (d = 0.5) was 204.

All questionnaires employed in this study were administered electronically, with data collection occurring between April 2020 and August 2020, coinciding with the COVID-19 outbreak in Europe. All questionnaires were distributed online through electronic questionnaires. The questionnaires were distributed based on the sample of students from the scientific research institutions in which the research team members worked. The questionnaire encompassed participants’ basic personal information and responses to the following three psychological scales: the Fear of Negative Appearance Evaluation Scale, The Eating Attitudes Test, and the Inventory of Physical Activity Objectives. The estimated time required to complete the questionnaire was approximately 15 min. The study protocol received approval from the Ethics Board for Research Projects at the Institute of Psychology, University of Gdansk, Poland (decision no. 33/2020), and was registered in the Protocol Registration and Results System at ClinicalTrials.gov; https://clinicaltrials.gov/ct2/show/NCT04432038 (accessed on 1 April 2020).

### 2.3. Instruments

#### 2.3.1. Fear of Negative Appearance Evaluation Scale

The Fear of Negative Appearance Evaluation Scale is an eight-item self-report instrument designed to measure concerns related to appearance evaluation [36]. This scale was developed by adapting items from the Brief Fear of Negative Evaluation Scale and introducing new items specifically addressing apprehension associated with negative appearance evaluative experiences. Six of the FNAES items (e.g., “I’m worried about what people think of my appearance”) were rated on a five-point Likert scale, ranging from disagree to agree. In the present study, the internal consistency of the FNAES, assessed through Cronbach’s alpha, was found to be 0.95. The value of McDonald’s Omega was found to be 0.83.

#### 2.3.2. The Eating Attitudes Test

The Eating Attitudes Test (EAT-26) stands as one of the most widely employed standardized self-report tools for identifying symptoms and concerns indicative of eating disorders [37]. Particularly valuable as a screening instrument, the EAT-26 is utilized in various settings, including high schools, colleges, and specialized groups like athletes, to assess the “risk” of eating disorders. Completion of the EAT-26 generates a “referral index” based on the following three criteria: (1) total score from EAT-26 responses; (2) answers to behavioral questions related to eating symptoms and weight loss; and (3) the individual’s body mass index (BMI) derived from height and weight measurements. Typically, a referral is recommended if a respondent scores “positively” or meets the designated “cut-off” scores on one or more criteria. Responses to questions 1 to 25 were scored as never/rarely/sometimes = 0, often = 1, usually = 2, or always = 3, while question 26 was scored in the opposite manner. Higher scores indicate more pronounced eating attitudes and behavioral concerns. A total score ≥ 20 suggests the potential presence of an eating disorder. In the current study, Cronbach’s alpha for the EAT-26 was 0.77. The value of McDonald’s Omega was found to be 0.72.

#### 2.3.3. Inventory of Physical Activity Objectives

The Inventory of Physical Activity Objectives scale (IPAO), introduced by Lipowski [38], serves as an effective method for assessing the motives behind physical activity and sports engagement, displaying robust psychometric properties. The IPAO is valuable for both scientific research and practical applications, aiding personal trainers in diagnosing motives for physical activity and sports participation. Comprising 15 questions, including basic personal information and exercise frequency, the IPAO assesses participants’ engagement in competitive sports (past and present) and their attitude toward passive involvement in sports. Responses to the 12 objectives are rated on a Likert-type scale (1–5). In the present study, the internal consistency of the IPAO, as measured by Cronbach’s alpha, was determined to be 0.90.

### 2.4. Data Analysis

The data analysis encompassed the utilization of SPSS 26.0 (IBM, Armonk, New York, NY, USA) and AMOS 26.0 (IBM, Armonk, New York, NY, USA), along with Matplotlib for visualization. Stage 1—Descriptive statistics were obtained by measuring the mean, standard deviation, skewness, and kurtosis of each variable in the research model. Differences and similarities among gender, age, and BMI values on appearance anxiety, eating attitudes, and physical activity goals were elucidated. Stage 2—An assessment of between-group differences for all variables was performed. For further statistical analysis, the normal distribution of the variables was examined using the Lilliefors test. The results of the test indicated that the variables did not satisfy the conditions for normal distribution. Therefore, a non-parametric test (Mann–Whitney U test) was used to measure the significance of differences in the variables by gender, age, and BMI. Stage 3—The strength of correlation among appearance anxiety, eating attitudes, and physical activity goals was measured using Spearman’s rank correlation coefficient. Stage 4—The strength of the relationship between the independent and dependent variables was measured using multiple regression analysis, and the mediating role of dietary attitudes was tested. Stage 5—A rank-based Kruskal–Wallis test was used to test the effects of two factors, age and BMI, on physical activity goals. A significance threshold of *p*-value < 0.05 was adopted to ascertain the statistical significance of the observed differences. BMI was obtained by dividing weight in kilograms by the square of height in meters.

## 3. Results

### 3.1. Descriptive Analysis

Table 1 presents the demographic and body mass index (BMI) data for the 644 adult participants. Among them, 359 individuals (55.7%) were aged 18 to 30, 143 individuals (22.2%) were aged 31 to 40, and 142 individuals (22.1%) were aged 41 or older. The gender distribution included 455 males and 189 females. The mean BMI of the cohort was 23.25, with a range of 14.52 to 33.5. Participants were categorized into underweight, normal, overweight, and obese groups based on their BMI values.

### 3.2. Relationship among Studied Variables

Table 2 shows the average values and the skewness and kurtosis of the variables used in this study.

The data in Table 3 and Figure 1 show that there is a significant correlation between FNAES and EAT-26 and IPAO, with correlation coefficients of 0.455 and 0.573, respectively. There is a significant correlation between EAT-26 and IPAO, with a correlation coefficient of 0.642.

Table 4, Figure 1 and Figure 2 presents the Pearson correlation coefficients among the various indicators. The data reveal that the Fear of Negative Appearance Evaluation Scale (FNAES) scores exhibited correlations with dieting (r = 0.38, *p* < 0.01), bulimia (r = 0.27, *p* < 0.01), oral control (r = −0.13, *p* < 0.01), and the Eating Attitudes Test-26 (EAT-26) total score (r = 0.41, *p* < 0.01). The FNAES also correlated with wellbeing (r = 0.52, *p* < 0.01) and decompression (r = 0.37, *p* < 0.01). Notably, there was a significant negative correlation between FNAES and oral control and no significant correlation with health, with the rest of the indicators demonstrating a significant positive correlation.

### 3.3. Fear of Negative Appearance, Eating Attitudes, and Physical Activity Objective Intergroup Differences

Table 5 presents the score disparities among survey respondents on the Fear of Negative Appearance Evaluation Scale (FNAES), Eating Attitudes Test-26 (EAT-26), and Inventory of Physical Activity Objectives (IPAO). Regarding age, individuals aged 41–50 years obtained the highest scores on the FNAES (16.06 ± 6.07), those aged 18–30 years had the highest scores on the EAT-26 (17.24 ± 13.4), and participants in the 41–50 years group achieved the highest scores across all three IPAO indicators, as follows: health, wellbeing, and decompression. Concerning BMI, the thin group recorded the highest score on the FNAES (16.23 ± 5.91), the obesity group had the highest total score on the EAT-26 (17.02 ± 13.5), and the normal group achieved the highest score in health (17.60 ± 2.6). The overweight group obtained the highest scores in wellbeing (15.01 ± 2.92) and decompression (16.03 ± 2.52).

### 3.4. Predictors of Physical Activity Objectives

The model was constructed with the Fear of Negative Appearance Evaluation Scale (FNAES) as the independent variable, the Inventory of Physical Activity Objectives (IPAO) as the dependent variable, and the Eating Attitudes Test-26 (EAT-26) as the mediating variable. A post-model analysis using AMOS 24.0 revealed a well-fitting model, as evidenced by the x^2^/*df* value of 1.328, which is less than 2, indicating a good fit between question items and an ideal model structure. Additionally, the root mean square error of approximation (RMSEA) value of 0.031, which is less than 0.05, signifies a satisfactory fit. The comparative fit index (CFI) value of 0.930 and the incremental fit index (IFI) value of 0.930, both exceeding 0.9, further affirm the well-fitted nature of the model.

Simultaneously, the standardized path coefficients of the variables incorporated into the structural equation demonstrated that FNAES scores significantly predicted EAT-26 scores (β = 0.25, *p* < 0.01), FNAES also significantly predicted IPAO scores (β = 0.27, *p* < 0.01), and EAT-26 significantly predicted IPAO scores (β = 0.17, *p* < 0.01) (see Figure 3).

The findings of the mediating effects, as outlined in Table 4, underscore the significant role of EAT-26 as a mediating variable. The direct impact of FNAES as an independent variable on the dependent variable IPAO is noteworthy. It can be inferred that EAT-26 exerts a completely positive mediating effect in the relationship between FNAES and IPAO, with the mediating effect accounting for 78.3% of the total effect (see Table 6).

The Table 7 and Figure 4’s data analysis presents the results of a two-way analysis of variance (ANOVA) investigating the interaction effect of age and body mass index (BMI) on the dependent variable. The interaction term, Age × BMI, showed a significant effect, with a significant F value of 21.471 (*p* < 0.000 **), indicating that the combined effect of age and BMI on IPAO exceeded their individual effects. The R-squared value (R^2^) was 0.576, and the adjusted R-squared value was 0.566, indicating that the model accounted for approximately 57.6% of the variability in the dependent variable, indicating moderate explanatory power.

These findings highlight the critical role of age and body mass index and their interaction in elucidating the variability observed in the dependent variable. They emphasize the need to consider the joint influence of multiple factors when studying complex relationships within a research domain.

## 4. Discussion

This study aimed to explore the intricate interplay among the Fear of Negative Appearance Evaluation Scale (FNAES), dietary attitudes, and the Inventory of Physical Activity Objectives (IPAO). Subsequently, additional investigation delved into discerning the mediating role of dietary attitudes in the relationship between FNAES and IPAO. Firstly, the positive effect of FNAES on the association between dietary attitudes and IPAO proposed for Hypothesis 1 was fully confirmed through research. A number of previous studies have suggested that negative appearance ratings significantly influenced individuals’ eating attitudes and were strongly associated with unhealthy eating behaviors [39]. Further research has also found that individuals with lower body image flexibility (i.e., higher worry and anxiety about appearance) are more likely to exhibit disordered eating behaviors. Negative appearance appraisal fears have been identified as an important factor in eating attitude change [40]. Another study exploring the relationship among body dissatisfaction, social media use, and physical activity levels among college students showed that negative appearance ratings were strongly associated with physical activity goals and that individuals coped with appearance anxiety by increasing physical activity [41]. Collectively, this literature suggests that the fear of negative appearance evaluation (FNAES) has a significant positive effect on the relationship between eating attitudes (EAT-26) and individual physical activity goals (IPAO).

Furthermore, the results of this study confirm the second hypothesis, affirming that dietary attitudes play a fully positive mediating role in the dynamic relationship between FNAES and IPAO. Several studies conducted over the past three years have provided evidence for this dynamic interaction. A survey has found that people with higher levels of body dissatisfaction are more likely to develop eating disorders, which can affect their physical activity levels [42]. It shows that dietary attitude has a significant mediating effect between FNAES and IPAO. Further evidence comes from a study of the relationship among body image, dietary attitudes, and physical activity among young adult women in South Africa. Research highlights that body dissatisfaction and the desire to control weight through eating behaviors significantly influence physical activity levels [43]. Additionally, another recent study focused on the impact of COVID-19-related distress on body image and eating behaviors in adolescents and young adults. Research has found that increased body image concerns during the pandemic have led to disordered eating behaviors that impact physical activity goals [44]. These studies support dietary attitudes as an important mediating factor between FNAES and IPAO. The complex interplay among the fear of negative appearance, dietary attitudes, and physical activity goals is highlighted, emphasizing the importance of addressing body image issues and dietary behaviors to promote healthier physical activity patterns.

Furthermore, the data collected from this study showed significant differences in FNAES, dietary attitudes, and IPAO across age and BMI categories, proving Hypothesis 3. Several recent studies have confirmed the impact of age stage and body mass index group on physical activity goals. Studies have shown that both age and body mass index significantly affect physical activity levels and goals, emphasizing the need for tailored interventions. This relationship is influenced by a variety of factors, including self-perception, physical ability, and socioeconomic conditions [45,46,47].

The comprehensive dataset revealed a positive influence of FNAES on the association between dietary attitudes and IPAO. Moreover, this study’s outcomes substantiated the initial hypothesis, affirming that dietary attitudes exert a completely positive mediating effect in the dynamic between FNAES and IPAO. Additionally, the data gleaned from this study unveiled noteworthy differences in FNAES, dietary attitudes, and IPAO across age and BMI categories. The findings derived from this study provide substantial support and enrichment to the existing research exploring the relationship between the Fear of Negative Appearance Evaluation Scale (FNAES) and dietary attitudes. Concerning FNAES scores, the results revealed a significant gender difference, with females scoring notably higher than males. This suggests that females harbor a heightened fear of negative appearance evaluations, aligning with the outcomes of several previous studies [48]. Numerous investigations consistently demonstrate that females assign greater importance to appearance and body image compared to males, with appearance evaluations significantly influencing women’s self-perception. Particularly pronounced effects are observed among individuals deviating from culturally prescribed beauty standards, as evidenced in this study among both obese and thin women. The research also illuminated noteworthy differences in FNAES scores across BMI subgroups.

The FNAES, measuring unique facets of body image, eating disorders, and depression, exhibited significant correlations with body image, eating attitudes, and mood levels [49,50]. Specifically, in the group of obese women, the fear of a negative appearance appraisal tended to correlate with the emergence of eating disorder situations [51]. Notably, this study did not identify a significant role of age in the fear of a negative appearance appraisal, with no discernible difference in fear scores across age groups. This finding diverges from some scholars’ studies, with some researchers positing that as age increases, changes in body size and appearance lead to a more negative self-evaluation [52,53]. This age-related increase in the fear of negative appearance evaluations is often emphasized in the middle-aged and older female population, attributing it to factors such as wrinkles and skin laxity [53,54]. The lack of a significant effect of age on the fear of negative appearance evaluations may be attributed to several factors. Social media platforms are a major source of appearance-based anxiety and are widely used across all age groups. Cultural norms and media portrayals often emphasize the importance of appearance and influence people of all ages [55]. The fear of negative judgment is a fundamental human issue that is tied to social acceptance and self-esteem [56]. In summary, the fear of negative appearance judgments is influenced by a combination of social, cultural, psychological, and media factors that affect individuals, regardless of their age. This generalization suggests that interventions aimed at reducing appearance-related anxiety should consider these wider influences, rather than focusing solely on age-specific strategies.

In the investigation of dietary attitudes, notable distinctions were found only among different age groups, specifically in the dieting subscale, in which significant differences were observed. Conversely, other demographic characteristics, encompassing distinct gender and BMI coefficient groups, did not exhibit substantial differences. Although females scored marginally higher than males in both dieting and oral control, as well as the total score, these differences did not attain statistical significance. This outcome diverges from the findings of certain prior studies, which commonly assert that females are more predisposed to eating disorders and are at a higher risk of experiencing food binge eating disorders compared to males. This inclination is often attributed to women’s proclivity for managing their body shape and appearance more intensively than men. Excessive management in this regard can lead to mood changes and eating disorder symptoms triggered by anxiety and stress [57].

Existing research on eating disorders has predominantly concentrated on the adolescent population. Therefore, further exploration of the mechanisms and psychological causes underlying the prevalence of eating disorders in diverse age groups is a promising avenue, particularly in the context of cross-sectional studies [57,58]. Noteworthy is the contextual backdrop of the COVID-19 pandemic within which this study was conducted. The pandemic imposed centralized and uniform restrictions on people’s behavioral activities, significantly influencing the eating behaviors of individuals across various ages and genders in their daily lives. The COVID-19 lockdown compelled men and women of all ages to stay at home, substantially elevating the likelihood of eating disorders, particularly among men. Several studies have highlighted the severe public health and mental health consequences of the COVID-19 pandemic [59]. Dietary restrictions or compensatory behaviors were accentuated during the epidemic [59], potentially exacerbating symptoms in individuals with eating disorders [60]. Proposing that a large-scale public health response possesses the potential to significantly modulate eating behaviors emerges as a valuable area for future research.

Regarding the Inventory of Physical Activity Objectives (IPAO) findings, notable distinctions emerged in physical behavioral activity goals among various age groups, particularly in the health and wellbeing subscales, with no significant differences in decompression. The prevailing physical activity goals leaned more toward health and wellbeing in the 40+ age group, suggesting that individuals in middle-aged and older demographics are inclined toward enhancing both their physical and mental health through exercise. However, no significant differences in decompression were noted across age groups. In contrast, significant differences in health, wellbeing, and depression were observed among BMI groups. This implies that individuals with different body types and weights harbor diverse motivations for engaging in specific physical activities. Overweight individuals exhibited the highest scores in health and depression, indicating a broader spectrum of motivations within this demographic [61,62].

A plethora of studies has consistently demonstrated a significant positive correlation between physical activity and mental health. The motivation for physical behavioral activity can influence the frequency of individuals’ participation, subsequently impacting participants’ mental health levels [63]. Simultaneously, this study asserts that the motivation to participate in physical behavioral activities holds more instructive value than the exercise itself. Participation motivation research primarily delves into intrinsic motivation, whereas exercise motivation predominantly explores extrinsic factors, such as appearance and weight management. Research focusing on intrinsic motivation is better poised to offer substantial guidance on the content and structure of physical activity. A study also showed that motivations to engage in physical activity can be used in interventions to increase physical activity participation by matching an individual’s primary motivations to the types of physical activity associated with these motivations in a large sample [64].

This study contributes empirical evidence indicating that the Fear of Negative Appearance Evaluation Scale (FNAES) exerts a direct positive influence on the Inventory of Physical Activity Objectives (IPAO), with dietary attitudes playing an indirect moderating role in this relationship. Additionally, this study revealed a noteworthy positive correlation between FNAES and eating attitudes. It is widely acknowledged that a prevalent and efficacious method for mitigating appearance anxiety typically involves dietary control, often resulting in irregular eating habits. Numerous studies exploring the connection between FNAES and eating behaviors have consistently concluded that FNAES serves as a superior predictor of eating attitudes and behaviors compared to other body image variables [65]. Specifically, women exhibited a particularly significant positive correlation between FNAES and eating attitudes. Our study further observed that heightened levels of fear related to negative body image appraisal positively correlated with dieting and bulimia behaviors within the realm of eating attitudes. Conversely, a significant negative correlation was identified for oral control. A cross-sectional investigation among female university students affirmed that both appearance comparisons and negative appraisals significantly influenced eating disorders [66]. Thus, it is plausible to infer that mitigating negative body image evaluations may contribute to a reduction in symptoms associated with eating disorders [67].

Research has consistently identified correlations between the Fear of Negative Appearance Evaluation Scale (FNAES) and goals related to physical behavioral activity. Numerous studies have indicated that dissatisfaction with body image often drives individuals to enhance their self-image through engagement in physical activity. Physical activity, in turn, indirectly contributes to reduced levels of psychological distress by fostering positive evaluations of one’s appearance. These studies suggest an indirect relationship between physical activity and psychological distress through appearance evaluation, a phenomenon observed in both males and females. Notably, the strength of this indirect effect tends to be more pronounced in females than in males. Furthermore, gender differences have been identified in the prevalence of psychological distress, investment in personal appearance, and satisfaction with one’s own appearance. Studies consistently report that girls exhibit higher levels of distress and invest more in appearance, coupled with lower satisfaction with their own appearance compared to boys. Notably, the most robust correlation exists between appearance evaluation and psychological distress, observed consistently in both females and males [68]. This study uniquely explores the impact of FNAES on physical activity goals, revealing significant positive associations between FNAES and goals related to wellbeing and stress reduction, which is a novel contribution to the existing literature. The findings suggest that a heightened fear of a negative body evaluation corresponds to an increased inclination to engage in physical activity for the purposes of wellbeing and decompression. This inclination may be attributed to the classification of FNAES within the realm of psychological stress, leading individuals to prioritize psychological stress relief when selecting physical activity goals.

The present study is subject to several noteworthy limitations. Firstly, the participants exclusively comprised Polish citizens, introducing potential variability in negative body image evaluations, dietary attitudes, and physical activity goals across cultural backgrounds. Therefore, caution is warranted in extending the applicability of these results to populations from other countries with distinct cultural contexts, necessitating further verification. Additionally, this study adopted a cross-sectional design, precluding the accurate derivation of longitudinal conclusions about the examined variables. Future investigations should comprehensively consider the perceptions of diverse age groups concerning negative appearance ratings and physical activity goals. Also, being limited by the sampling technique of this study led to imperfect sample characteristics and other factors. The results of this study only support this sample. The applicability in a wide range of samples and in populations with other characteristics needs further validation. Furthermore, it is crucial to acknowledge that this study was conducted during the COVID-19 pandemic, and consequently, the conclusions specifically pertain to outcomes within a unique public health context. Generalizing these conclusions to more typical, everyday situations necessitates additional research. The analysis of variance based on the BMI index should be considered representative of the current sample because of the large differences in the sex ratios of men and women. Extra attention should be paid to this issue in future studies. Despite these limitations, the current study offers detailed and valuable insights into the intricate relationship among negative appearance ratings, physical activity goals, and dietary attitudes within the Polish population. Undoubtedly, there remains a need for in-depth analyses of the relationship between dietary attitudes and body image, as well as physical activity, aiming to enhance mental and physical health, particularly in the context of significant public safety and health events.

## 5. Conclusions

The main aim of this study was to investigate the impact of the fear of a negative appearance evaluation on eating attitudes and physical activity goals during COVID-19, which was characterized by unique lockdown measures that significantly altered people’s eating and physical activity behaviors. The goal of this study was to explore the strong relationship among appearance anxiety, eating attitudes, and physical activity goals and explore whether dietary attitudes mediate the relationship between FNAES and IPAO. The current study confirms that the fear of a negative appearance evaluation can positively predict individuals’ dietary status and physical activity goals during COVID-19, with a significant positive correlation. This suggests that changing attitudes toward appearance assessment may impact individuals’ daily physical activity status, particularly under lockdown. Furthermore, studies have shown that the fear of a negative appearance evaluation not only directly affects physical activity goals but also exhibits a significant full mediating effect on eating attitudes. Therefore, changing attitudes toward appearance evaluation may help improve eating disorder status and thereby improve the quality of physical activity. The main theoretical implication of this study is to provide valuable insights into the relationship among appearance evaluations, eating attitudes, and physical activity. It emphasizes the importance of psychological factors in these behavioral patterns.

## Figures and Tables

**Figure 1 nutrients-16-02065-f001:**
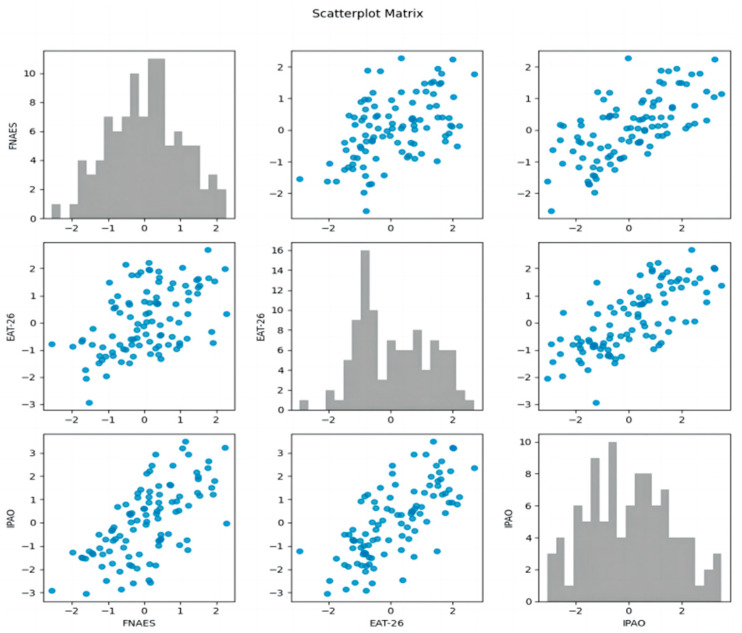
The scatter plot matrix shows the pairing relationship among the three variables, FNAES, EAT-26, and IPAO. Shown on the diagonal is a histogram for each variable, with scatterplots between variables shown elsewhere.

**Figure 2 nutrients-16-02065-f002:**
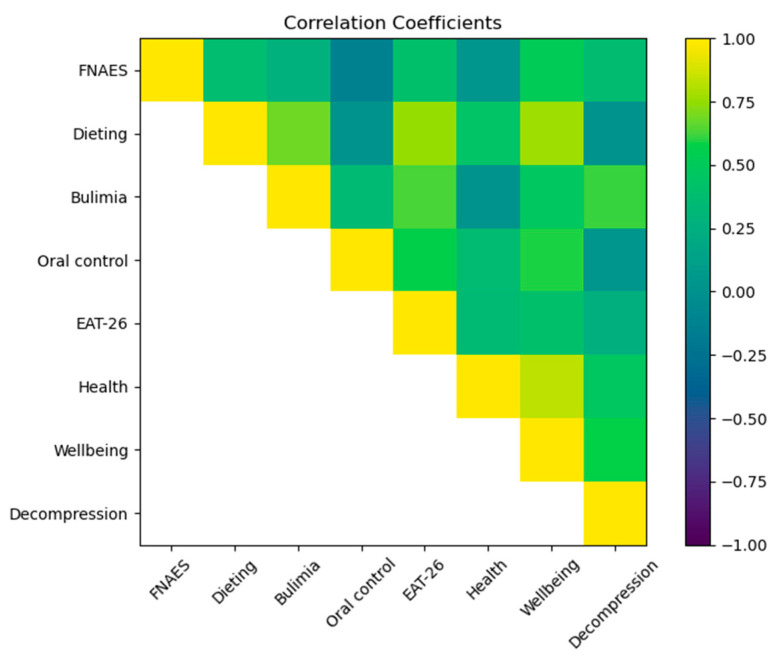
Heatmap of correlation among FNAES, EAT-26, and IPAO.

**Figure 3 nutrients-16-02065-f003:**
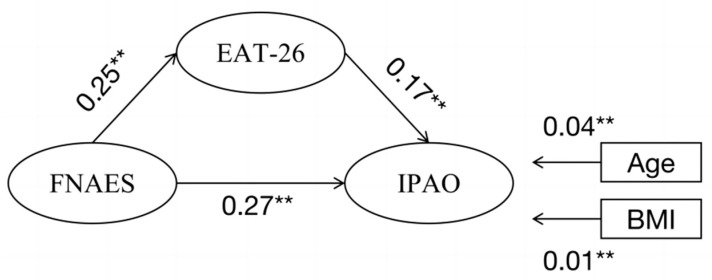
Path analysis of the relationship among FNAES, EAT-26, and IPAO of participants. Values represent the significant standardized regression coefficients. ** *p* < 0.01.

**Figure 4 nutrients-16-02065-f004:**
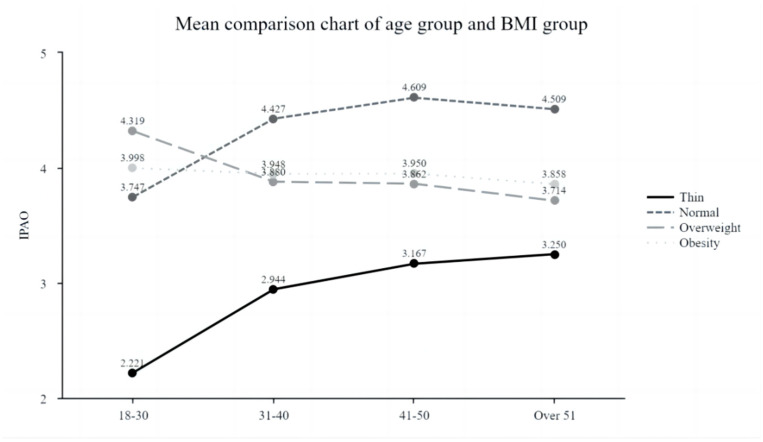
Mean comparison chart of age group and BMI group.

**Table 1 nutrients-16-02065-t001:** Description of participants.

Demographic	Value
Total Participants	644
Male	455
Female	189
Mean BMI	23.25
Minimum BMI	14.52
Maximum BMI	33.5
BMI Category: ≤18.4	58
BMI Category: 18.5–23.9	346
BMI Category: 24.0–27.9	154
BMI Category: ≥28.0	86

**Table 2 nutrients-16-02065-t002:** Mean values, standard deviation, skewness, and kurtosis of the variables.

Variables	Mean ± SD	95% CI (LL)	95% CI (UL)	IQR	K	S	(CV)
FNAES	2.575 ± 1.044	2.494	2.656	1.667	−0.845	0.216	40.538%
EAT-26	4.520 ± 1.010	4.466	4.575	0.923	18.209	−3.279	22.345%
IPAO	3.909 ± 0.750	3.851	3.967	1.000	1.019	−0.906	19.186%

SD = standard deviation; S = skewness; K = kurtosis.

**Table 3 nutrients-16-02065-t003:** Bivariate Pearson correlation coefficient.

	FNAES	EAT-26	IPAO
FNAES	1		
EAT-26	0.455 *	1	
IPAO	0.573 *	0.642 *	1

* *p* < 0.05.

**Table 4 nutrients-16-02065-t004:** Correlation among FNAES, EAT-26, and IPAO.

	FNAES	Dieting	Bulimia	Oral Control	Eat-26	Health	Wellbeing	Decompression
FNAES	1							
Dieting	0.38 **	1						
Bulimia	0.27 **	0.68 **	1					
Oral control	−0.13 **	0.03	0.35 **	1				
Eat-26	0.41 **	0.75 **	0.63 **	0.57 **	1			
Health	0.04	0.43 **	0.02	0.36 **	0.35 **	1		
Wellbeing	0.52 **	0.77 **	0.47 **	0.60 **	0.41 **	0.829 **	1	
Decompression	0.37 **	0.02	0.61 **	0.04	0.25	0.471 ***	0.581 **	1

** *p* < 0.01, *** *p* < 0.001.

**Table 5 nutrients-16-02065-t005:** Age and BMI groups in FNAES, EAT-26, and IPAO (differences among the groups).

	Category (N)	FNAES	EAT-26				IPAO		
			Dieting	Bulimia	Oral Control	Total Score	Health	Wellbeing	Decompression
Age	18–30 years	15.23 ± 6.40	4.67 ± 0.83	2.59 ± 0.53	2.68 ± 0.31	17.24 ± 13.40	15.36 ± 3.80	15.36 ± 3.80	15.47 ± 3.57
	31–40 years	15.62 ± 6.01	3.70 ± 0.50	2.45 ± 0.58	2.32 ± 0.72	14.78 ± 12.12	17.64 ± 3.10	17.64 ± 3.10	16.05 ± 2.67
	41–50 years	16.06 ± 6.07	4.39 ± 0.26	2.03 ± 0.38	2.36 ± 0.60	15.34 ± 10.52	18.38 ± 2.51	18.38 ± 2.51	16.15 ± 3.07
	Over 51 years	15.56 ± 6.28	3.98 ± 0.40	2.16 ± 1.22	2.79 ± 0.32	14.89 ± 11.97	18.03 ± 2.82	18.03 ± 2.82	15.92 ± 2.92
	H-value	0.443	2.80 **	1.44	1.47	1.76	30.50 ***	30.52 ***	1.66
	η^2^	0.002	0.01	0.007	0.007	0.008	0.125	0.10	0.008
BMI	≤18.4	16.23 ± 5.91	3.68 ± 0.26	2.26 ± 0.47	2.57 ± 0.03	14.47 ± 12.00	8.65 ± 2.45	8.70 ± 2.09	12.00 ± 3.54
	18.5–23.9 Normal	15.56 ± 6.21	4.53 ± 0.57	2.50 ± 0.43	2.54 ± 0.07	16.61 ± 12.32	17.60 ± 2.61	15.17 ± 2.49	16.09 ± 3.00
	24.0–27.9 Overweight	14.88 ± 6.41	4.21 ± 0.83	2.40 ± 0.58	2.42 ± 0.11	15.63 ± 13.28	17.04 ± 2.96	15.77 ± 3.64	16.20 ± 3.24
	≥28.0	15.47 ± 6.63	4.53 ± 0.96	2.50 ± 0.43	2.97 ± 0.38	17.02 ± 13.53	16.50 ± 3.65	15.01 ± 2.92	16.03 ± 2.52
	F value	0.754	1.08	0.188	1.26	0.716	20.50 ***	21.60 ***	32.92 ***
	η^2^	0.002	0.005	0.001	0.006	0.003	0.49	0.32	0.13

** *p* < 0.01 *** *p* < 0.001.

**Table 6 nutrients-16-02065-t006:** Bootstrap analysis of the mediation effect size and significance test of FNAES in IPAO and EAT-26.

Path	Standardized Effect Size (Effect)	Standard Error (Boot SE)	Effect Size (%)	95% CI LL UL
FNAES→IPAO (Direct effect)	0.27 **	0.15	31.10	(0.27, 0.53)
FNAES→EAT-26→IPAO (Mediation effect)	0.36 **	0.26	78.3	(0.37, 0.65)
FNAES→IPAO (Total effect)	0.54 **	0.39	100	(0.33, 0.70)

CI confidence interval, LL lower limit, UL upper limit. ** *p* < 0.01.

**Table 7 nutrients-16-02065-t007:** Age and BMI Kruskal–Wallis analysis of IPAO values.

	Sum of Square	*df*	Mean Square	F	*p*
Age	6.352	3	2.117	57.498	0.000 **
BMI	32.405	3	10.802	44.245	0.000 **
Age × BMI	47.175	9	5.242	21.471	0.000 **

R^2^ = 0.576 (Adjusted R^2^ = 0.566), ** *p* < 0.01.

## Data Availability

The datasets generated during and analyzed during the current study are available from Li, jianye (2024), “Navigating Body Image Anxiety: Unraveling the Interplay of Appearance Evaluation Fear, Eating Attitudes, and Physical Activity Objectives amidst COVID-19”, Mendeley Data, V2, doi: 10.17632/yttdbxcsm2.2.

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
