# Peer review of "Predictive Model of the Relationship between Appearance, Eating Attitudes, and Physical Activity Behavior in Young People amid COVID-19"

_nutrients, 2024, doi:10.3390/nu16132065_

Round 1
Reviewer 1 Report
Comments and Suggestions for Authors
Thank the authors for their efforts to expand scientific knowledge about psychological aspects so important for the mental health and quality of life of young people, such as physical appearance, eating behavior, and physical exercise behavior.
Specific comments
Recommended Title: "Predictive model of the relationship between between appearance, eating attitudes, and physical activity behavior in young people amidst COVID-19"
Abstract
- The conclusion described in the abstract is not based on the main conclusions derived from the obtained results.
- It's suggested to modify the keywords to avoid being the same as those used in the manuscript title.
- The characteristics of the sample should be included (mean values and standard deviation of age, sex, percentage of men and women and level of education).
Introduction
• The theoretical foundation of this section should support the understanding of behavior concerning the variables under study.
• Recent studies (last 3-4 years) of randomized controlled trials (RCTs) and/or quasi-experimental designs (with control groups and pre-post intervention designs) on the analyzed variables in the target population should be presented, such as physical appearance and dietary behavior in the young population.
• A theoretical justification for the proposed structural regression model is required. This involves arguing about the theories that explain the relationship between physical appearance, eating disorders, and the motives behind physical activity and sports engagement.
Materials and Methods
Research Design
- The method should start with a description of the study's design, for example: "The design was quantitative, quasi-experimental, and cross-sectional (Hernández Sampieri, 2018)". The employed methodology (quantitative and/or qualitative) and the type of design according to the methodology should be specified (e.g., pre-post intervention design).
- The research design needs to be described with the corresponding bibliographic reference.
Participants
- Due to the age heterogeneity and origin of the sample, it's crucial to provide the inclusion and exclusion criteria for participation in the study should be described.
- The selected sampling technique for the study must be indicated, as well as information on the statistical test conducted to calculate the sample size prior to data collection.
Measures or Instruments
· Authors are recommended to report the value of McDonald's Omega for the reliability of the FNAES and EAT tests.
Procedure
- Specific details about the message sent through social networks to recruit the sample and how the participants were informed about the study's objective
- How was the sample recruited?
- When and how were the questionnaires administered to them?
Data Analysis
- It's crucial to mention that non-parametric inferential statistical tests were used due to the non-normal distribution of the sample.
- The Kolmogorov-Smirnov test is not provided. Concerning this test, assuming the mean and population variance as known, which in most cases is impossible, makes the test very conservative and less powerful. To address this issue, a modification of the Kolmogorov-Smirnov test known as the Lilliefors test was developed. The Lilliefors test assumes that the mean and variance are unknown, being specially developed to test normality.
Results
- Authors are advised to organize the description of this section as follows: 1) First, report the mean values, standard deviation, skewness, and kurtosis of the variables under study (for each factor of the tests). 2) Second, present the table and the figure of bivariate correlations of the studied variables. 3) Third, describe the values of ANOVA or non-parametric test for the analyzed variables. 4) Fourth, describe the procedure followed to carry out the measurement model and the structural regression model, as well as the goodness-of-fit indices data for both models. To adequately describe the results of the SEM model, please refer to the following article: [Ortiz, Manuel S., & Fernández-Pera, Montserrat. (2018). Modelo de Ecuaciones Estructurales: Una guía para ciencias médicas y ciencias de la salud. Terapia psicológica, 36(1), 51-57. https://dx.doi.org/10.4067/s0718-48082017000300047].
- Specify in the table footer all acronyms related to variables and statistical data. For example: Note. M = Mean; SD = Standard desviation; S = skewness; K = kurtosis; SP = Spelling Performance; * p < .05; ** p < .01
Important note regarding the Results
It is necessary to define which triggers exactly influence physical activity behavior, that is, define the variable "Triggers" (e.g.: peers, teachers' opinions, family opinions, social networks, etc.).
Discussion
- It is necessary to report the confirmation or refutation of the hypotheses raised and compare the results with current studies (last 3-4 years) on the study variables.
- It is crucial to argue about not finding statistically significant data between the study variables.
- Additionally, it would be helpful to provide a better explanation of the study's limitations. For example: the sampling technique, the quasi-experimental and biased nature of the sample treatment, the characteristics of the control group, etc.
Conclusions
The conclusions must be directly related to the objectives, which should be written in the Introduction.
References
· The font style (“font”) of the references does not match the font style used throughout the manuscript.
· Review that all citations are referenced according to the journal's required format. The publication year should be bolded. For example: Meiklejohn, B.N.D.; Ryan, L.; Palermo, C. A Systematic Review of the Impact of Multi-Strategy Nutrition Education Programs on Health and Nutrition of Adolescents. J. Nutr. Educ. Behav. 2016, 48, 631-646. https://doi.org/10.1016/j.jneb.2016.07.015
Finally, I wish the authors all the best in continuing this line of research.
Decision: The reviewer considers that the suggested changes in the specific comments are necessary for the manuscript to be publishable.
Author Response
|
Response to Reviewer 1 Comments Manuscript ID: nutrients-2982697
|
||
|
1. Summary |
|
|
|
Thank you very much for taking the time to review this manuscript.The suggestions you have made are very relevant and actually improve the completeness and quality of the article. Once again, thank you for your efforts. Please find the detailed responses below and the corresponding revisions/corrections highlighted/in track changes in the re-submitted files. |
||
- Point-by-point response to Comments and Suggestions for Authors
Comments 1:Recommended Title: "Predictive model of the relationship between appearance, eating attitudes, and physical activity behavior in young people amid COVID-19"
Response 1: Thank you for your suggestions. We think the title you suggested is very suitable for this article, making the title of the article more attractive. Therefore, the title of the article has been changed according to your suggestion. (See the title on the first page)
Comments 2:The conclusion described in the abstract is not based on the main conclusions derived from the obtained results.
Response 2: Thank you for your suggestions. We have modified the conclusion part in the abstract part based on the obtained results. (See page 1, lines 15-25)
Comments 3:It's suggested to modify the keywords to avoid being the same as those used in the manuscript title.
Response 3: Based on the suggestions you provided, we have reorganized the keywords to avoid duplication of words in the article title.
Comments 4:The characteristics of the sample should be included (mean values and standard deviation of age, sex, percentage of men and women and level of education).
Introduction
Response 4: The age and proportion characteristics of the sample have been added to the summary based on your suggestions. (See page 1, lines 12-13)
Comments 5:The theoretical foundation of this section should support the understanding of behavior concerning the variables under study.Recent studies (last 3-4 years) of randomized controlled trials (RCTs) and/or quasi-experimental designs (with control groups and pre-post intervention designs) on the analyzed variables in the target population should be presented, such as physical appearance and dietary behavior in the young population.A theoretical justification for the proposed structural regression model is required. This involves arguing about the theories that explain the relationship between physical appearance, eating disorders, and the motives behind physical activity and sports engagement.
Response 5: In response to your comments above, we have added a description of the relevant theoretical basis and theoretical framework and literature references in the introduction section. The suggestions you gave greatly improved the theoretical basis of this study. Thank you so much. (Please attend lines 41-46 on page 2, lines 96-98 on page 3, and lines 109-146 on page 3)
Comments 6:The method should start with a description of the study's design, for example: "The design was quantitative, quasi-experimental, and cross-sectional (Hernández Sampieri, 2018)". The employed methodology (quantitative and/or qualitative) and the type of design according to the methodology should be specified (e.g., pre-post intervention design).
Response 6: Based on your suggestion, the research type of the study is quantitative, correlational and descriptive cross-sectional. (See page 5, lines 158-169)
Comments 7:The research design needs to be described with the corresponding bibliographic reference.
Response 7: Based on your suggestion, the literature citation identification has been added to the research design section. (Please go to page 5, line 159)
Comments 8:Due to the age heterogeneity and origin of the sample, it's crucial to provide the inclusion and exclusion criteria for participation in the study should be described.
Response 8: Based on your suggestions, we have added a description of the inclusion and exclusion criteria for participating in the study. (Please go to page 5, lines 175-179)
Comments 9:The selected sampling technique for the study must be indicated, as well as information on the statistical test conducted to calculate the sample size prior to data collection.
Response 9: Based on your suggestions, we have supplemented the sampling techniques selected for the study. and relevant information for calculating sample size prior to data collection. (Please go to page 5, line 175, lines 180-182)
Comments 10:Authors are recommended to report the value of McDonald's Omega for the reliability of the FNAES and EAT tests.
Response 10: Based on your suggestion, we have added the value of McDonald's Omega for the reliability of the FNAES and EAT tests. (Please go to page 6, line 205, line 222)
Comments 11:Specific details about the message sent through social networks to recruit the sample and how the participants were informed about the study's objective
How was the sample recruited?
Response 11: Based on your suggestion, we have provided additional explanations in the article. (Please go to page 6, lines 163-169)
Comments 12:When and how were the questionnaires administered to them?
Response 12: Based on your questions, we have provided additional explanations in the article. (Please go to page 6, lines 183-188)
Comments 13:It's crucial to mention that non-parametric inferential statistical tests were used due to the non-normal distribution of the sample.
Response 13: According to your reminder, the Lilliefors test has been supplemented in the article and tested to show that the sample is normally distributed. (Please go to page 7, lines 243-244)
Comments 14:The Kolmogorov-Smirnov test is not provided. Concerning this test, assuming the mean and population variance as known, which in most cases is impossible, makes the test very conservative and less powerful. To address this issue, a modification of the Kolmogorov-Smirnov test known as the Lilliefors test was developed. The Lilliefors test assumes that the mean and variance are unknown, being specially developed to test normality.
Response 14: Thank you for the reminder. We've added instructions for normality testing based on your suggestion. (Page 7 244-246)
Comments 15:Authors are advised to organize the description of this section as follows: 1) First, report the mean values, standard deviation, skewness, and kurtosis of the variables under study (for each factor of the tests). 2) Second, present the table and the figure of bivariate correlations of the studied variables. 3) Third, describe the values of ANOVA or non-parametric test for the analyzed variables. 4) Fourth, describe the procedure followed to carry out the measurement model and the structural regression model, as well as the goodness-of-fit indices data for both models. To adequately describe the results of the SEM model, please refer to the following article: [Ortiz, Manuel S., & Fernández-Pera, Montserrat. (2018). Terapia psicológica, 36(1), 51-57. https://dx.doi.org/10.4067/s0718-48082017000300047].
Response 15: Thank you for the order and structure of the data chart. We combined existing research and made adjustments based on your suggestions. Added description of each variable. At the same time, bivariate correlation tables and visualization diagrams have been added. (See page 8, line 269; Table 2, Table 3, figure1)
Comments 16:Specify in the table footer all acronyms related to variables and statistical data. For example: Note. M = Mean; SD = Standard desviation; S = skewness; K = kurtosis; SP = Spelling Performance; * p < .05; ** p < .01 .Important note regarding the Results
Response 16: Based on your suggestion, the table footer in the article was checked for all acronyms related to variables and statistics specified. (See page 8, line 270)
Comments 17:It is necessary to define which triggers exactly influence physical activity behavior, that is, define the variable "Triggers" (e.g.: peers, teachers' opinions, family opinions, social networks, etc.).
Response 17: Thank you for your suggestion, but I am sorry that I am a little confused about your suggestion and I hope you can answer it patiently.
Comments 18:It is necessary to report the confirmation or refutation of the hypotheses raised and compare the results with current studies (last 3-4 years) on the study variables.
Response 18: Thank you for your reminder. We have added relevant content to the discussion section. (Please go to page 14, lines 351-393)
Comments 19:It is crucial to argue about not finding statistically significant data between the study variables.
Response 19: Based on your suggestion, we have added a comparative discussion of variables with no significant differences in the discussion supplement. (Please go to page 16, lines 422-433)
Comments 20:Additionally, it would be helpful to provide a better explanation of the study's limitations. For example: the sampling technique, the quasi-experimental and biased nature of the sample treatment, the characteristics of the control group, etc.
Response 20: Based on your suggestions, we have added relevant content to the limitations section to improve the rigor of the research. (Please go to page 19, lines 540-550)
Comments 21:The conclusions must be directly related to the objectives, which should be written in the Introduction.
Response 20: In response to your suggestions, we have matched the research results with the objectives in the conclusion section. (Please go to page 19, lines 560-563)
Comments21:The font style (“font”) of the references does not match the font style used throughout the manuscript.Review that all citations are referenced according to the journal's required format. The publication year should be bolded. For example: Meiklejohn, B.N.D.; Ryan, L.; Palermo, C. A Systematic Review of the Impact of Multi-Strategy Nutrition Education Programs on Health and Nutrition of Adolescents. J. Nutr. Educ. Behav. 2016, 48, 631-646. https://doi.org/10.1016/j.jneb.2016.07.015
Response 21: Based on your suggestions, we have checked and adjusted the format of all references.
Reviewer 2 Report
Comments and Suggestions for Authors
Authors report data obtained from a cross-sectional study performed on a sample of 644 adult citizens form Poland aimed at evaluating the possible interplay of fear appearance evaluation, eating attitudes and physical activity during the onset of COVID-19 pandemic.
The topic had a great interest during the period characterized by the relevant restrictions in terms of social activities. Data obtained during such a unique moment (i.e. April-August 2020) appear hardly generalizable. This represents the main limitation of the study, adequately reported by authors in the appropriate section.
The paper has several conceptual and methodological flaws that further weakens the relevance of the results presented.
Major points
1. The Introduction section appear narrow, misleading and lacks adequate references. First, the concept of “fear of negative appearance” should adequately defined, with appropriate references. The perception of one’s appearance can influence eating attitudes and physical activity in healthy subjects. Authors should clarify whether they are investigating a pathological or a physiological construct. Secondly, as the sample is driven from general population, all the dimensions investigated in the study should be referred to non-clinical samples. Furthermore, most sentences appear neither accurate, nor specific and some are completely not acceptable and/or not sustained by literature (e.g. “Lukas further broke down the symptoms of eating disorders in his study, and he concluded the most common eating disorders including are anorexia nervosa, bulimia nervosa and binge eating disorder[13]” (Ref. 13 is wrong!); “Individuals afflicted with eating disorders frequently exhibit profound dissatisfaction with their physical form, often perceiving themselves as overweight or aesthetically unpleasing despite empirical evidence suggesting otherwise[14]”; “From this, it can be assumed that eating disorder symptoms can influence a person's physical behavior activity goals”; “A literature search revealed that there are fewer studies on diet and exercise for stress reduction, which is a key point to explore in depth in this study”).
Finally, given the above reported remarks, the main hypothesis and aims of the study are not sustained by the main body of the Introduction.
2. No power calculation, based on a primary endpoint, has been performed. This point is always crucial in cross sectional study but has a particular relevance since authors did not describe how subjects were enrolled.
The sample is largely composed by men (70.4 %). This point is surprisingly neglected. All data on gender differences are flawed by this disproportion (455 vs. 189) and considerations on data obtained in females are based on a small number of subjects.
In the first par. of Methods section are reported data that should be moved in the Results section.
3. The first par. of the Results section should be completely rewritten and Figure 1 is unnecessary. Data should be reported in a succinct manner, without comment (that should be reported in the discussion section).
Table 2. should not include gender comparisons and caption should be rewritten.
Comments on the Quality of English LanguageNone.
Author Response
|
Response to Reviewer 2 Comments Manuscript ID: nutrients-2982697
|
||
|
1. Summary |
|
|
|
Thank you very much for taking the time to review this manuscript.The suggestions you have made are very relevant and actually improve the completeness and quality of the article. Once again, thank you for your efforts. Please find the detailed responses below and the corresponding revisions/corrections highlighted/in track changes in the re-submitted files.
|
||
- 2. Point-by-point response to Comments and Suggestions for Authors
Comments 1: The Introduction section appear narrow, misleading and lacks adequate references. First, the concept of “fear of negative appearance” should adequately defined, with appropriate references. The perception of one’s appearance can influence eating attitudes and physical activity in healthy subjects. Authors should clarify whether they are investigating a pathological or a physiological construct. Secondly, as the sample is driven from general population, all the dimensions investigated in the study should be referred to non-clinical samples. Furthermore, most sentences appear neither accurate, nor specific and some are completely not acceptable and/or not sustained by literature (e.g. “Lukas further broke down the symptoms of eating disorders in his study, and he concluded the most common eating disorders including are anorexia nervosa, bulimia nervosa and binge eating disorder[13]” (Ref. 13 is wrong!); “Individuals afflicted with eating disorders frequently exhibit profound dissatisfaction with their physical form, often perceiving themselves as overweight or aesthetically unpleasing despite empirical evidence suggesting otherwise[14]”; “From this, it can be assumed that eating disorder symptoms can influence a person's physical behavior activity goals”; “A literature search revealed that there are fewer studies on diet and exercise for stress reduction, which is a key point to explore in depth in this study”).
Finally, given the above reported remarks, the main hypothesis and aims of the study are not sustained by the main body of the Introduction.
Response 1: Based on your valuable suggestions, we have reorganized and supplemented the introduction part. Firstly, the definition of fear of negative appearance is added, and secondly, the literature that you pointed out is not relevant is deleted. (See page 2, lines 41-46)
Comments 2. No power calculation, based on a primary endpoint, has been performed. This point is always crucial in cross sectional study but has a particular relevance since authors did not describe how subjects were enrolled.
The sample is largely composed by men (70.4 %). This point is surprisingly neglected. All data on gender differences are flawed by this disproportion (455 vs. 189) and considerations on data obtained in females are based on a small number of subjects.
In the first par. of Methods section are reported data that should be moved in the Results section.
Response 2: Based on your relevant suggestions, we have supplemented the article with the calculation of the power, as well as the sample sampling technique, inclusion criteria, and description of the recruitment method. and illustrates among the limitations the imbalance of the male-female sample. (Please go to page 5, lines 175-182, 185, 188; page 19, lines 558-561)
Comments 3. The first par. of the Results section should be completely rewritten and Figure 1 is unnecessary. Data should be reported in a succinct manner, without comment (that should be reported in the discussion section).
Table 2. should not include gender comparisons and caption should be rewritten.
Response 3: Thank you for your suggestion. Based on your suggestion, we have re-simplified the sentences in the first par. of the results section and removed redundant commentary sentences. (Please go to page 7, lines 259-264); Table2 has now been changed to table5, and the gender comparison has been deleted, and the caption has been modified. (See page 12, line 299)
Reviewer 3 Report
Comments and Suggestions for Authors
This is a clear and well-written manuscript evaluating the relationships between fear of negative appraisal of physical appearance, eating disorders, and physical activity in a sample of more than 640 Poland participants. The content is original, and the statistical analysis seems correct.
My contribution to further improvement of the manuscript concerns some aspects that should be supplemented or explained:
1. In the Introduction, it is necessary to discuss more the perception of physical appearance during COVID-19 based on the literature (see, for example, https://link.springer.com/article/10.1007/s10578-022-01364-1; 10.3390/healthcare11142101);
2. In the Methods, there is no mention of the traits Stature, and Weight needed to calculate BMI. If they were included in the questionnaires it should be specified;
3. BMI was calculated with the sexes combined. This is only acceptable if there are no significant differences between sexes and this should be stated;
4. At the end of the Discussion, at least two other points of weakness should be included: the numerical imbalance between males and females within the sample and the BMI data derived from self-reported data.
Minor concerns:
- You used Covid-19 and COVID-19 indifferently in the text. The acronym comes from COronaVIrus Disease 2019 and should always be indicated with capital letters (COVID-19).
- In the Introduction change “ their appearance is being scrutinized. and is…” to “their appearance is being scrutinized and is…”.
- - Change the title of paragraph 3. RESULT to 3. RESULTS.
- - In the Discussion change "motivations to engage in physical avtivity can be used in interventions" to "motivations to engage in physical activity can be used in interventions “.
Comments on the Quality of English Language
None
Author Response
|
Response to Reviewer 3 Comments Manuscript ID: nutrients-2982697
|
||
|
1. Summary |
|
|
|
Thank you very much for taking the time to review this manuscript.The suggestions you have made are very relevant and actually improve the completeness and quality of the article. Once again, thank you for your efforts. Please find the detailed responses below and the corresponding revisions/corrections highlighted/in track changes in the re-submitted files. |
||
2. Point-by-point response to Comments and Suggestions for Authors
Comments 1:1. In the Introduction, it is necessary to discuss more the perception of physical appearance during COVID-19 based on the literature (see, for example, https://link.springer.com/article/10.1007/s10578-022-01364 -1; 10.3390/healthcare11142101);
Response 1: Based on your suggestion, we have added the literature on people's appearance perceptions during Covid-19 to the introduction section. (Please go to lines 52-54 on page 2)
Comments 2: In the Methods, there is no mention of the traits Stature, and Weight needed to calculate BMI. If they were included in the questionnaires it should be specified;
Response 2: Based on your suggestions, we have added the BMI calculation standard in methods. (Please go to page 7, lines 255-256)
Comments 3: BMI was calculated with the sexes combined. This is only acceptable if there are no significant differences between sexes and this should be stated;
Response 3: Thank you for your valuable suggestions, we have added relevant explanations in the limitations section. And in the next research, we will pay special attention to the issue of male-female ratio. (Please go to page 19, lines 540-550)
Comments 4: At the end of the Discussion, at least two other points of weakness should be included: the numerical imbalance between males and females within the sample and the BMI data derived from self-reported data.
Response 4: Based on your suggestion, we have added a description of this issue to the Limitations Statement in the Discussion section. (Please go to page 19, lines 540-550)
Comments 5: You used Covid-19 and COVID-19 indifferently in the text. The acronym comes from COronaVIrus Disease 2019 and should always be indicated with capital letters (COVID-19).
Response 5: Thank you for your careful discovery, we have replaced 25 covid-19 with COVID-19.
Comments 6: In the Introduction change “their appearance is being scrutinized. and is…” to “their appearance is being scrutinized and is…”.
Response 6: Based on your suggestion, we have made replacements in the text. (Please go to page 2, line 48)
Comments 7: Change the title of paragraph 3. RESULT to 3. RESULTS.
Response 7: Based on your prompts, we have corrected it. (Please go to page 7, line 257)
Comments 8: In the Discussion change "motivations to engage in physical avtivity can be used in interventions" to "motivations to engage in physical activity can be used in interventions “.
Response 8: Based on your suggestion, we have completed the replacement. (Please go to page 17, line 484)
Reviewer 4 Report
Comments and Suggestions for Authors
Thank you for the interesting paper. Here are some comments for the authors to improve the paper.
Introduction:
Please add that before reaching extreme eating disorders, people usually start with emotional eating, especially when the overall context is a crisis like a global pandemic. See: https://doi.org/10.3390/foods13091347
The studies cited: specify the demographic it was conducted among (youth, students, adults, and which country).
It is essential to emphasize the significance of the current study and its scientific contribution beyond previous studies.
Methods:
Move the description of the sample to the Results section.
The research procedure is missing - it is not clear how the questionnaires were distributed and to whom. Which software was used, how many participants entered, and what sampling method was used? Were there versions of the questionnaires translated into Polish? Were they translated for the current research? How were the questionnaires validated?
Results:
Please add sub-headings.
Figures 1, 2, and 3 are redundant. The information is clearly presented in the tables.
Discussion:
The authors wrote: "A plethora of studies have consistently demonstrated." However, they brought only one example [41]. It repeats itself throughout the discussion. The authors, in general, should add more literature to the discussion and provide a deeper explanation of findings that do not align with previous research.
In the study's limitations, it is necessary to include the bias due to the overrepresentation of males. Additionally, the sample is a convenience sample that is not representative. Moreover, this is a cross-sectional study. Therefore, causality cannot be established, and the observed effects may not be related to the pandemic.
Conclusions:
I disagree with the sentence, "The current research confirms that COVID-19 has indeed changed people's daily lives and has had long-term and far-reaching impacts." While the study was conducted during the COVID-19 pandemic, these effects are likely temporary and unrelated. Moreover, the participants did not complete questionnaires before and after the pandemic or were not asked if it influenced their behavior.
Author Response
|
Response to Reviewer 4 Comments Manuscript ID: nutrients-2982697
|
||
|
1. Summary |
|
|
|
Thank you very much for taking the time to review this manuscript.The suggestions you have made are very relevant and actually improve the completeness and quality of the article. Once again, thank you for your efforts. Please find the detailed responses below and the corresponding revisions/corrections highlighted/in track changes in the re-submitted files. |
||
- Point-by-point response to Comments and Suggestions for Authors
Comments 1: Please add that before reaching extreme eating disorders, people usually start with emotional eating, especially when the overall context is a crisis like a global pandemic. See: https://doi.org/10.3390/foods13091347
Response 1: Based on your suggestion, we have added references to relevant literature in the text. (Please go to page 2, lines 58-60)
Comments 2:The studies cited: specify the demographic it was conducted among (youth, students, adults, and which country).
Response 2:Following your suggestion, we have added demographi information to the description of the relevant cited literature. (Please see page 2, line 52; page 3, lines 99, 111.)
Comments 3:It is essential to emphasize the significance of the current study and its scientific contribution beyond previous studies.
Response 3: Based on your suggestions, we have added relevant research significance and modest contributions to the article. (Please go to page 4, pages 142-146)
Comments 4:Move the description of the sample to the Results section.
Response 4: Based on your suggestions, we have moved the relevant content to the rusults section. (See page 7, pages 259-264)
Comments 5:The research procedure is missing - it is not clear how the questionnaires were distributed and to whom. Which software was used, how many participants entered, and what sampling method was used? Were there versions of the questionnaires translated into Polish? Were they translated for the current research? How were the questionnaires validated?
Response 5: Thank you for your valuable suggestions. We have provided additional explanations on the above content in the article. (Please go to page 5, lines 158-169)
Comments 6:Please add sub-headings.
Response 6: Based on your suggestion, we have added subtitles for each part of the study in the results section. (Please go to page 7, line 258; page 8, line 266; page 11, line 286; page 11, line 301)
Comments 7:Figures 1, 2, and 3 are redundant. The information is clearly presented in the tables.
Response 7: Thank you for your suggestion. We have deleted figure 1. We hope to decide whether to retain the remaining figures after further communication with you. Visualization of tabular data may provide a more intuitive display of the data. Ultimately we will respect your suggestions and make final adjustments.
Comments 8:The authors wrote: "A plethora of studies have consistently demonstrated." However, they brought only one example [41]. It repeats itself throughout the discussion. The authors, in general, should add more literature to the discussion and provide a deeper explanation of findings that do not align with previous research.
Response 8: Based on your valuable suggestions, we have added relevant content and literature to the discussion section. Especially the discussion of inconsistent findings (see page 16, lines 422-433)
Comments 9:In the study's limitations, it is necessary to include the bias due to the overrepresentation of males. Additionally, the sample is a convenience sample that is not representative. Moreover, this is a cross-sectional study. Therefore, causality cannot be established, and the observed effects may not be related to the pandemic.
Response 9: Based on your suggestions, we have added relevant instructions to the limitations section. Thank you again for your rigorous approach. (Please go to page 19, lines 540-550)
Comments 10:I disagree with the sentence, "The current research confirms that COVID-19 has indeed changed people's daily lives and has had long-term and far-reaching impacts." While the study was conducted during the COVID-19 pandemic, these effects are likely temporary and unrelated. Moreover, the participants did not complete questionnaires before and after the pandemic or were not asked if it influenced their behavior.
Response 10: Thank you for the suggestions. We have removed this statement.
Round 2
Reviewer 2 Report
Comments and Suggestions for Authors
The authors have deeply revised the manuscript following all the remarks raised. The paper appears significantly improved.
Author Response
Dear reviewers, thank you for your high-quality suggestions and valuable time. Thank you for your recognition of the article.
Reviewer 4 Report
Comments and Suggestions for Authors
The authors addressed all the comments and improved the paper. It is now possible to accept for publication.
Author Response
Dear reviewer, your suggestions have greatly improved the quality of the article, and thank you for your final approval.